# Online HIV Self-Testing (HIVST) Dissemination by an Australian Community Peer HIV Organisation: A Scalable Way to Increase Access to Testing, Particularly for Suboptimal Testers

**DOI:** 10.3390/ijerph182111252

**Published:** 2021-10-26

**Authors:** Sara Fiona Elisabeth Bell, Jime Lemoire, Joseph Debattista, Andrew M. Redmond, Glen Driver, Izriel Durkin, Luke Coffey, Melissa Warner, Chris Howard, Owain David Williams, Charles F. Gilks, Judith Ann Dean

**Affiliations:** 1School of Public Health, Faculty of Medicine, The University of Queensland, Herston 4006, Australia; sara.bell@uq.edu.au (S.F.E.B.); odwdub@gmail.com (O.D.W.); c.gilks@uq.edu.au (C.F.G.); 2Queensland Positive People, East Brisbane 4169, Australia; lemoire@me.com (J.L.); andrew.redmond@health.qld.gov.au (A.M.R.); glendriver@outlook.com (G.D.); idurkin@qpp.org.au (I.D.); lcoffey@qpp.org.au (L.C.); mwarner@qpp.org.au (M.W.); CHoward@qpp.org.au (C.H.); 3Metro North Public Health Unit, Metro North Hospital and Health Service, Windsor 4030, Australia; joseph.debattista@health.qld.gov.au; 4Royal Brisbane and Women’s Hospital, Metro North Hospital and Health Service, Herston 4006, Australia

**Keywords:** peer-led, intervention, feasibility, HIV, testing, HIV self-testing (HIVST), men who have sex with men (MSM), gay, Australia

## Abstract

HIV self-testing (HIVST) introduces opportunities for screening in non-conventional settings, and addresses known testing barriers. This study involved the development and evaluation of a free online HIVST dissemination service hosted by a peer-led, community-based organisation with on-site, peer-facilitated HIV testing, and established referral and support programs for people newly diagnosed with HIV to determine whether this model was feasible and acceptable for engaging MSM, particularly among infrequent and naive HIV-testers, or those living in remote and rural areas. Between December 2016 and April 2018, 927 kits were ordered by 794 individuals, the majority of whom were men who have sex with men (MSM) (62%; 494), having condomless sex (50%; 392), or living outside a major city (38%; 305). Very few (5%; 39) sought the available pre-test peer contact, despite 45% (353) being naive HIV-testers. This study demonstrates that online HIVST dissemination is acceptable and feasible for engaging at-risk suboptimal testers, including those unwilling to test elsewhere (19%; 47/225). With half (50%; 403) unwilling to buy a kit, our study suggests that HIVST will need to be subsidized (cost-neutral to users) to enhance population coverage and access.

## 1. Introduction

Access to HIV treatment is a vital element for the control and elimination of HIV [1]. Early testing and diagnosis facilitate timely access to HIV treatment and ongoing care, resulting in improved health outcomes and prevention of onward transmission. Improving access to and uptake of HIV testing is a critical strategy for reducing the number of people living with undiagnosed HIV [2]. Male-to-male sex remains the predominant risk of exposure in Australia [3]. A 2010 study from Queensland—Australia’s third largest state—identified that 20% of men who have sex with men (MSM) living with HIV were unaware of their HIV status [4]. More recent national data from 2018 estimate that the proportion of Australian MSM living with undiagnosed HIV has fallen to 9% [5]. This suggests that Australia is moving in the right direction, towards achieving 95% of people living with HIV (PLHIV) being aware of their diagnosis [3]. However, despite an ongoing focus on targeted HIV testing among MSM, modelling conducted in 2015 estimated that undiagnosed HIV among MSM accounted for 59% of new HIV infections in 2015 [6], and Queensland notification data reported that 68% of new HIV diagnoses in 2018 were among people with no documented HIV test result in the previous 12 months [7]. This suboptimal testing frequency suggests that barriers to HIV testing remain in Australia [8].

Barriers to HIV testing have been well documented. Structural barriers include geographical distance between client and clinic [9,10], clinic opening times [9,11], difficulty in booking appointments [9], and lack of testing for additional sexually transmissible infections (STI) [9]. Individual barriers include low perceived risk [2,11], fear of a positive test result [2,11,12] , fear of or being subject to stigma and discrimination [9,12,13], lower age [2], less education [2], non-identification as same-sex-attracted [2], inability to speak the language of the resident country [2], and limited financial resources [2]. On the supply side, barriers include lack of healthcare provider knowledge [11] and poor systems of confidentiality [13].

Technological innovation through point-of-care testing (POCT) has introduced new opportunities for HIV testing in non-conventional settings, whilst offering advantages such as convenience, reduced result waiting times, and reduced barriers of perceived stigma from services [14,15]. Peer-based community POCT models of service facilitate access to HIV testing among MSM—particularly among naive HIV-testers (people who have never been tested before) [16]. HIV self-testing (HIVST) provides convenience and privacy for the person testing [17], and offers additional opportunities to address identified barriers to testing [12,15,18,19]—especially when appropriately targeted to at-risk populations [20] and combined with peer-based support [21]. Mathematical modelling suggests that HIVST will increase access to and uptake of HIV testing if used to complement existing conventional and rapid testing practices [22,23], at very little cost [24]. A recent Australian randomized controlled trial demonstrated that the availability of HIVST for MSM doubled HIV testing frequency overall, and resulted in a fourfold increase in testing among non-recent HIV testers [25]. Some people have expressed concerns about the lack of pre- and post-testing information and support afforded to individuals when using HIVST [15]. However, a recent systematic review and meta-analysis comparing HIVST to standard testing noted that HIVST increases testing uptake and frequency for MSM and trans people, without negative effects on linkage to care, STI testing, condom use, or social harm [26].

HIVST is yet to be fully embedded into Australia’s suite of existing HIV testing options, despite HIVST being approved for market in Australia from December 2018 [27]. Understanding of how to implement innovative dissemination models will support HIVST implementation science. Such models need to include mechanisms of choice for accessing peer and clinician support pre- and post-testing, along with pathways facilitating early linkage to care and support of those with a reactive HIVST result to confirmatory testing [15,28]. A particular strength of HIVST is the capacity for internet-based service delivery, especially among people known to have embraced online technologies [29] and wanting discrete choice [30]. Online approaches to access HIVST offer practical means to expand existing HIV and public health programs [31,32] to target hard-to-reach populations and infrequent and naïve HIV-testers [15]. Additionally, online distribution has been reported as the preferred means of accessing HIVST kits [15,30], and allows for innovative ways for enabling linkage to peer support [21].

Here, we report the results of an integrated model of ordering free HIVST kits online through a website hosted by an established peer-led community organisation for PLHIV. This implementation study set out to determine whether an online HIVST distribution service model is feasible and acceptable for engaging MSM in Queensland, Australia, and whether it increased access to HIV testing among infrequent and naive HIV-testers and MSM living in remote and rural areas. Of particular interest was how people engaged with the service model, along with their preferences for HIVST compared to other modes of testing, pre-test information and contact preferences, their willingness to pay for HIVST, and their experience of using an HIVST kit and navigating linkage to care. The outcomes of this study are particularly relevant in the Australian context, where significant regional health disparities exist, and will provide an evidence-informed opportunity to assess the present online-ordered HIVST service model implemented in Australia and globally.

## 2. Materials and Methods

### 2.1. Study Design

A single-site implementation study with an embedded mixed-methods monitoring and evaluation (M&E) [33] was conducted by a multidisciplinary team of peers, clinicians, and researchers from The University of Queensland (UQ) and Queensland Positive People (QPP)—a peer-led HIV-community-based organisation that has been providing on-site peer-facilitated asymptomatic HIV and syphilis POCT since 2014 [16] and, more recently, molecular POCT for *Chlamydia trachomatis* (CT) and *Neisseria gonorrhoeae* (NG) [34,35,36].

The study involved the development of a study registration page hosted on the established QPP website, which linked to an online order system for the HIVST kit. The webpage also included links to resources relating to HIV, testing, and living positive, along with referral and support services for people newly diagnosed with HIV. QPP Peer Test Facilitators (PTF) coordinated the receipt and postage of HIVST kit orders. They also provided pre- and post-test information as requested by participants, and facilitated referral where necessary.

The embedded M&E component, designed to determine whether such a model is feasible and acceptable for engaging MSM—particularly among infrequent and naive HIV-testers, or those living in remote and rural areas, included quantitative online survey data collected at two points in time (pre- and post-HIVST) combined with qualitative data obtained via a series of interviews collected throughout the study [33,37]. UQ coordinated and conducted the M&E component and analysed de-identified data. The study protocol has been previously described in detail [38]. This paper presents the results of the quantitative online survey data collected pre- and post-dissemination of the kits.

### 2.2. Participants and Recruitment

The study, while targeting MSM, was open to participants of all genders and sexual orientations, aged 18 years or older, and living in Queensland, Australia. Requesting delivery of the HIVST kit to a postcode outside of Queensland was the primary exclusion criterion. A combination of five active recruitment strategies described in more depth previously [38] were employed, including:Geotargeted advertising via mobile online banner advertisements on two gay-identifying apps (i.e., Grindr and Squirt) and Facebook;Promotion on non-gay-identifying classified advertisement websites (e.g., Craigslist);Dissemination of study information in gay press and by HIV and peer organisation websites in Queensland;Search engine optimization (SEO);Word of mouth and respondent-driven sampling.

### 2.3. Intervention

#### 2.3.1. Informed Consent and Online Ordering

On registration via the HIVST online ordering webpage, participants were asked to provide informed consent via completion of a brief survey collecting data on the participant characteristics, reasons for HIV testing, reasons for the use of HIVST, previous experience, and willingness to pay for HIVST. They were also provided access to a downloadable participant information sheet (PIS).

Participants were then offered the choice of three pre-test information options: (1) HIVST kit posted with no PTF contact or oral pre-test information; (2) HIVST kit posted following PTF-initiated contact with participant where oral pre-test information is given and guidance on kit use is discussed; or (3) HIVST kit posted following participant-initiated contact with PTF for oral pre-test information and guidance on kit use. Participants were also asked to opt in to receive three monthly testing reminders via phone, email, or text messaging, and to be contacted by a PTF via phone two weeks after HIVST kit distribution for a brief structured interview and to receive a link to a post-test survey via text message.

#### 2.3.2. HIVST Distribution

The HIVST kits were distributed via Australia Post in a non-identifying package. In addition to the OraQuick ADVANCE^®^ HIV-1/2 Test [39] used for this study, the kit included:Instructions on how to perform and read the test (written and diagrammatic);Information on pre-and post-exposure prophylaxis and the benefits of regular testing;Information to assist with linkage to care in the event of a reactive result, including contact details of clinical and support services.

#### 2.3.3. Post-Test Follow-Up

A brief structured telephone interview designed to ascertain whether the HIVST kit was received, whether the test was conducted, whether there were any issues in performing or reading the HIVST, the result of the HIVST, and to offer the participant the opportunity to ask questions, was conducted by the PTF two weeks after mailing of the HIVST kit. If participants reported a reactive result, they were supported in accessing HIV management and care services for confirmatory HIV testing as per QPP’s standard operating procedures (SOPs). Participants would also be provided with additional assistance through QPP’s existing PLHIV peer navigation system, or same-day consultation with a specialist medical provider.

Participants were also sent a link to a post-test survey at this point to explore the use of the online ordering system, usefulness of the information provided, use of the HIVST kit, reflections on pre-and post-test information provided by the PTF, willingness to pay, and HIV and STI testing practices.

### 2.4. Ethical Issues, Safety, and Adverse Events

QPP, the peer-led PLHIV organisation that implemented the trial, has a history of offering innovative community-based HIV testing with a respectful client-centred approach [16]. The research was undertaken with the informed consent (implied by the voluntary completion of the online registration form) of all participants.

Due to the nature of HIV testing occurring in private and/or at home by the participant, anticipated risks were mitigated by a range of strategies. These included provision of HIV information via an evidence-based infographic on using the HIVST device designed specifically for this study, as well as contact details for support organisations. Participants were also provided access to PTFs, working in accordance with the medical-officer-approved SOPs, trained in a broad range of social and medical aspects of HIV, HIV stigma, use of the HIV self and POCT, the pre-test information, and in discussing HIV POCT results and follow-up of clients whose HIV POCT is reactive. A strict risk management framework addressing potential physical and psychological stresses to participants, PTFs, and researchers was embedded into the model. Should the result of the HIVST be reactive, the participant was supported in accessing HIV management and care services for confirmatory HIV testing and provided additional assistance through QPP’s existing peer navigation system, as outlined in the organisation’s SOPs.

Risk was further mitigated by the meaningful involvement of affected communities and peers throughout the study process. The HIVST dissemination and follow-up model was also based on a current service delivery model developed by QPP in collaboration with peer test providers and people living with and affected by HIV, who have accrued several years’ experience and expertise in the delivery of HIV testing and post-test follow-up.

### 2.5. Data Analysis

Data analysis conducted included summary statistics, univariate analysis, and bivariate analysis, appropriate to the variable types, using analytic software (IBM Corp., Released 2016. IBM SPSS^®^ Statistics for Windows, Version 24.0. Armonk, NY, USA). Statistically significant associations (*p* < 0.05) between groups were assessed with chi-squared tests. The primary outcome of the study—the number of occasions of service arising from an online HIVST ordering service—was assessed via demographic and testing history information gathered upon study enrolment. Secondary outcomes (recruitment sources, service user profiles including sexual and testing behaviour and location, pre- and post-test information choice, reasons for selecting HIVST, willingness to pay for HIVST) were measured by descriptive statistics (counts, percentages), and data were analysed to determine whether the target audience—specifically, MSM populations, infrequent and non-testers, and those living in regional and remote Queensland—were engaged.

## 3. Results

Between December 2016 and April 2018, 794 individuals received 927 HIVST kits (794 first orders, 133 repeat orders). A further 136 orders from interstate and international requests were received during the study period, but were unfilled, as they did not meet the study criteria. Of the 794 first orders, 62% (494) were MSM, 85.3% (677) were cis males, 48% (380) were aged 20–29 years, 38% (305) lived outside a major city, 59% (469) recorded an income of less than AUD50,000/annum, and 10% (81) had no Medicare (National Health Insurance) card. A schematic representation of the study implementation flow and the outcomes of the first ordered HIVST kits (*n* = 794) is provided in Figure 1.

Increasing uptake of the service in terms of total orders and repeat orders was observed over the study period (Figure 2).

### 3.1. Recruitment

Of the total number of individual first orders (794), 29% (237) were recruited through internet searches, 29% (237) through Facebook, 24% (199) through word of mouth, 11% (90) through gay dating apps, 4.7% (39) through HIV organisations and websites, 1.3% (11) through the ‘gay media’, 0.6% (5) through respondent-driven sampling, and 0.2% (2) not specified.

### 3.2. Behaviour

Most first orders (69%; 550) reported having 1–5 partners in the previous six months, with a further 15% (121) reporting 6–10 partners. Having had more than 10 partners in the previous six months was reported by 8.3% (65) of participants. The odds of identifying as MSM (odds ratio (OR) 2.4; 95% confidence interval (CI) 1.7–3.4) or placing a repeat HIVST order (OR 1.6; 95% CI 1.1–2.4) were increased for participants reporting 6 or more partners in the previous 6 months compared to those with 1–5 partners.

Half (50%; 392) of all first orders received indicated condomless sex as a motivation for HIV testing; 40% (309) of orders indicated never testing for HIV before as the reason for seeking HIV testing, whilst 35% (273) of orders indicated testing as part of regular screening practices. Thirty-three (3%) orders cited condomless sex while taking HIV pre-exposure prophylaxis (PrEP), and 1.0% (8) reported shared injecting equipment. Of the repeat orders (133), 65% (87) were requested for routine screening, and 47% (63) due to recent condomless sex.

### 3.3. Engagement of Population Groups

With respect to the study’s objectives of exploring the feasibility and success of the model to engage target subpopulations, the following was observed.

#### 3.3.1. MSM

Of the 794 individuals who ordered kits, 62% (494) were MSM: 50% (397) reported sex with men only, 12% (95) reported sex with men and women, and 0.3% (2) reported being bisexual transgender men. Of repeat orders (133), 79% (105) were from MSM, including 62% (83) reporting sex with men only, and 17% (22) for men who have sex with men and women. Identifying as MSM was significantly associated with ordering more than one HIVST kit during the study period (15.823, *p* < 0.001). The odds of reporting MSM sexual behaviour (OR 2.5, 95% CI 1.6–3.9) were increased for repeat HIVST orders.

#### 3.3.2. Remote and Rural Populations

Of first orders (794), 62% (489) of participants lived in a major Queensland city, as defined by the Australian Bureau of Statistics (ABS). Participants from Inner Regional Queensland, Outer Regional Queensland, Remote Queensland, and Very Remote Queensland accounted for 20% (161), 17% (131), 0.6% (5), and 1.0% (8) of orders, respectively.

The number of HIVST kits mailed to locations outside major Queensland cities increased during the study period. This was particularly evident for those in Inner and Outer Regional Queensland and correlated with the introduction of geotargeted sponsored Facebook advertising in the last month of study quarter (SQ) 3, with a focus on the Inner and Outer Regional areas of Queensland (Figure 3).

#### 3.3.3. Infrequent and Naive HIV-Testers

No previous HIV test was reported by 45% (353) of first order participants, with a further 30% (235) reporting that their last HIV test was more than 12 months prior. Almost one-third (31%; 123) of the men who only had sex with men reported having never tested for HIV, compared with 59% (56) of men who had sex with men and women (MSMW) (24.356, *p* < 0.001). The odds of ever having had an HIV test were decreased by 30% for MSMW (OR 0.3, 95% CI 0.2–0.5) compared to MSM. Living outside of a major Queensland city was not associated with having had an HIV test previously (0.142, *p* = 0.707) or with time since last HIV test (2.819, *p* = 0.244).

### 3.4. Preferred Delivery of Pre- and Post-Test Information Provided within the Service Model

The vast majority (95%; 755) of first order participants did not wish to have pre-test contact with a PTF prior to receiving the HIVST kit; a further 2.5% (20) of participants chose to contact a PTF at QPP themselves for pre-test information before the HIVST was posted, and 2.4% (19) requested that a QPP PTF contact them before postage of the HIVST kit. The choice not to engage with a PTF pre-test was significantly associated with no previous HIV testing (3.833, *p* = 0.05).

Post-test surveys were partially completed on 246 occasions. Of those responding, only 24% (30/124) agreed or strongly agreed that they *“found it beneficial to talk with a peer health worker over the phone **before** completing the test…”*. More respondents (47%; 81/173) agreed or strongly agreed that they *“found it beneficial to talk with a peer health worker over the phone **after** completing the test”*.

### 3.5. Reasons for Selecting HIVST

Reasons for choosing to test for HIV using the HIVST kit did not vary by first and repeat orders. Overall, participants reported that the reasons they chose to test for HIV via the online HIVST project were due to convenience (79%; 726), not wanting to wait for results (44%; 402), not wanting to talk about sex with anyone (33%; 298), not having time to go elsewhere for a test (29%; 268), and fear of stigma (22%; 205). Lack of local HIV testing services was reported for 7.2% (66) of orders.

Of the first orders, a few participants (8.8%; 70) reported having used an HIVST previously. Previous HIVST use was significantly associated with being born overseas (14.435, *p* < 0.001); however, ‘not having a Medicare card’ was not associated with prior use of HIVSTs (311, *p* = 0.577).

Of the 245 participants responding to the post-test online survey question “*I prefer to test for HIV at home than in a clinic*”, 85% (208) agreed or strongly agreed.

Of the 190 first order participants who responded to the survey question “*Where would you have tested if HIVST was not available?”,* 21% (40) reported they would not have tested elsewhere. Of these 40 participants, 55% (22) had not been tested previously, and 23% (9) had not been tested for more than 12 months. Not previously testing for HIV was significantly associated with respondents who would not have tested elsewhere without this project (5.032, *p* = 0.025). The odds of ever having had an HIV test were decreased by 40% for those who reported that they would not have tested elsewhere (OR 0.4; 95%CI 0.2–0.9) compared to those who would have.

During the study period, 95 (14%) participants ordered two or more HIVST kits (range 2–7, median 2, interquartile range (IQR) 2–3 HIVST kits). Of repeat orders (133), 32% (43) were within 1–90 days of their previous order, 39% (52) were within 91–180 days of their previous order, and 29% (38) of orders were received more than 180 days after the previous order.

### 3.6. Follow-Up of Results

Post-test peer worker contact with participants was achieved for 52% (485) of HIVST orders. Despite three attempts to contact participants, 48% (440) were unable to be contacted. Failed follow-up contact was not associated with participants who cited “*not wanting to talk to anybody*” as a reason for self-testing (0.305, *p* = 0.581), nor with previous HIV testing experience (0.013, *p* = 0.910).

### 3.7. Management of Reactive Results

One participant reported a reactive result during the study period; with the support of the resources provided in the HIVST kit, the participant successfully self-navigated their way to confirmatory HIV testing and linkage with an HIV healthcare provider prior to the two-week follow-up call. During the follow-up telephone call, the PTF was able to link the participant with the QPP Peer Navigation Program.

### 3.8. Willingness to Pay

Of first orders, 51% (403/794) of participants indicated that they would not be willing to pay for an HIVST, whereas 45% (354/794) indicated a price they would pay for an HIVST (AUD$5–100, median AUD$20, IQR AUD$15–30). Of repeat orders, a smaller proportion of orders (37%; 50/133) indicated that they would not be willing to pay for an HIVST, and whilst the median price point was still AUD$20, the IQR was higher (AUD$20–30). Willingness to pay was not significantly associated with those who would not have tested elsewhere (1.136, *p* = 0.286).

## 4. Discussion

This project sought to improve access to HIV testing for suboptimal testers through a novel online postal program for HIVST kits embedded within an established peer-led rapid HIV testing service with formalised links to support for people newly diagnosed with HIV. Disseminating HIVST kits via an online-ordered HIVST website run by a peer-led HIV community organisation offers a means of support and advice to those users who want it, while also serving to engage users who wanted to access the HIVST without direct service contact, as evidenced by 21% (40) of first-time users stating that they would not have tested for HIV elsewhere. With 45% (353) of the 794 individuals ordering HIVST kits reporting no previous HIV testing, and a further 30% (235) reporting that their last HIV test was more than 12 months prior, this study also suggests that the implemented HIVST model successfully engaged those who had never been tested for HIV. With 64% of all orders from among MSM, our findings suggest that this model of HIVST dissemination is an acceptable and feasible means of reaching infrequent and naive HIV-testers among this priority group.

The availability of HIV testing via online ordering and postal delivery should, in principle, allow all people equity of access to HIV testing, regardless of their geographic location of residence. A small number of participants resided in outer regional (4.9%) or remote parts of Queensland (1.9%), and while this intervention offers only limited data, we would suggest that HIVST offers an opportunity to increase testing in remote and rural Australia, especially when coupled with strategies such as the implementation of geotargeted Facebook advertising to increase awareness of this testing option. The increased numbers of tests ordered by people living in regional remote areas that occurred in response to geotargeted Facebook advertising also indicate that the use of social media is an effective method of engaging with residents of regional areas. Poor local access to testing services was cited by 7.2% (66) of orders as a reason for ordering an HIVST. With the majority (63%) of tests ordered by urban ‘city-dwelling’ or inner regional Queensland (20%) residents, our study also highlights the fact that barriers to HIV testing in Queensland extend beyond geographical distance to testing services.

Stigmas around HIV, sex, and marginalised sexual identities are acknowledged barriers to HIV testing [40]. Stigma is not limited solely to regional and remote areas of Queensland, but may exist in communities, or be directed by or perceived from health services. More than one in five (22%; 205) survey respondents indicated that the HIVST orders were because of fear of stigma. In addition, 33% (298) of survey respondents indicated not wanting to talk about sex with anyone. For a potential tester, ordering online removes the necessity of having direct contact with a practitioner. Further research is needed into the influence that these perceived and enacted experiences of stigma may have on linkage to care following a reactive HIVST.

Flexible models of testing such as HIVST potentially provide for greater convenience, given that such services are not restricted to locations fixed by time and space. Within our study, most orders (79.4%; 726) cited convenience as their reason for using HIVST, with a further 29.3% (268) of orders being due to service users not having time to go elsewhere for a test.

With respect to the issue of cost and potential financial barriers, this project provided HIVST kits free of charge. Sale of HIVST kits is likely to have limited capacity to meet the need of those who want self-testing as, consistent with previous Australian [15,28] and international [41] research, many (50.8%) of the study participants indicated that they would not be willing to pay for an HIVST kit. In 2019, the TGA approved the Atomo Diagnostics™ rapid diagnostic finger-prick test in Australia for use as a self-test. This provides the opportunity for Australian MSM to order a safe and effective HIVST online for AUD$25 plus AUD$15 postage (total AUD40). However, with AUD20 being the median cost that participants were willing to pay in this study, and only 9.8% (78) of first orders willing to pay AUD$40 or more, this suggests that the current commercial market-driven HIVST distribution model may be a barrier to uptake, and that in order to be effective, HIVST programs would require external funding or subsidization to enhance population coverage and access.

Overwhelmingly, pre-test support was not requested by MSM ordering an HIVST in this study. The authors were cognizant of the concerns surrounding the lack of pre-test discussion with HIV self-testing [42,43,44], so the study model was purposively developed to investigate whether people choosing to use an HIVST would choose to have contact with a PTF prior to testing or after using the kit. The HIVST kit was accompanied by a leaflet titled *“Getting prepared to start your home HIV self-test”*, which provided access to comprehensive and accurate pre-test information as well as links to other HIV testing information, including clear written and diagrammatic instructions on how to use the kit. This may have negated the need for participants to seek peer support [28]. The “*Getting prepared*” leaflet, developed specifically for this study, was added to the HIVST Implementation Toolkit available through the AIDSFree HIV Testing Services Knowledge Base, on request from the World Health Organization (WHO) [45].

Despite the very low uptake of pre-test information, almost half of those surveyed agreed or strongly agreed that it was beneficial to talk with a PTF over the phone after completing the test. This supports the importance of embedding a mechanism whereby people can choose to have contact with a peer in online dissemination models. In this study, a PTF contacted participants 2 weeks after mailing of the HIVST kit, to ascertain whether the HIVST kit had been received and provide an opportunity to commence dialogue.

With respect to clinical practice and providing pathways for linkage to medical care and peer support, this study has directly demonstrated the feasibility for the model to provide state-wide access via a single hub. The community-based organisation that hosted and implemented the project, QPP, was able to offer support to participants state-wide via telephone contact and the provision of contacts local to participants for confirmatory HIV testing or other sexual health services, as required. QPP PTFs were ideally placed to seamlessly link people with the QPP PLHIV Peer Navigation Program (Life+) for those recently diagnosed with HIV and, thus, to link people with a confirmed HIV diagnosis into early treatment and care with HIV specialist health services. This model could be scaled up nationally and tailored to meet the needs of other at-risk populations. Each Australian jurisdiction could utilise the Queensland model, while providing jurisdiction-specific support and linkages. A single national centre of co-ordination could also be considered, with the necessary and appropriate local referral information available. However, this would potentially be more problematic in terms of sources and proportions of funding. Alternatively, smaller states and territories could partner with a larger state’s service.

### Strengths and Limitations

A noted strength of this study is that it was co-designed and led by peers in collaboration with a multidisciplinary team of clinicians, researchers, and PLHIV. Partnerships with PLHIV and those at risk are central to understanding how HIVST kits can be added to the existing suite of HIV testing modalities to facilitate testing uptake among those not testing or accessing other testing options. This peer-supported model also has health economic value, reducing clinical service demands and redirecting clinicians’ time and expertise to complex case management [24]. This study does, however, have some limitations. Firstly, even with the use of several recruitment strategies designed to target infrequent and naive HIV-testers, there is the potential that those recruited may be people who are already linked with HIV- and gay-community-driven social media information. This study also does not provide access to comprehensive STI testing; however, the model provides an acceptable framework to include self-test kits for other STIs when they become available. The low post-test response rate also limits the ability to evaluate participants’ experience of using an HIVST kit and navigating linkage to care. More research is needed to inform linkage to clinical care of people screening in non-clinical settings.

## 5. Conclusions

This study suggests that online access to HIVST is a feasible and acceptable testing option for MSM, particularly for engaging those who have never been tested or who would not have been tested through other methods. This well-received model of HIVST distribution—implemented by a peer PLHIV community organisation with established links to HIV testing information, referral, and support for people newly diagnosed with HIV—can ameliorate psychosocial, financial, and geographical barriers to HIV testing, whilst providing support for those wishing to test for HIV. However, with less than 10% of participants willing to pay the retail price, this study suggests that market-driven online HIVST dissemination is unlikely to increase HIV testing rates. Effective HIVST programs would require external funding or subsidization to enhance population coverage and access. The registering of an HIVST, and the subsequent availability of the device on the open market, is not an absolute panacea to ameliorate barriers to HIV testing access. This study demonstrates that the provision of HIVST kits free of charge by a recognised and trusted PLHIV community organisation, via an online ordering system with embedded clinical governance and duty of care to PLHIV, is a successful method of engagement. The provision of free HIVST kits, as a public health intervention to those at high risk of HIV, adds to the suite of HIV testing options, irrespective of the financial, geographic, and psychosocial barriers affecting an individual’s access to HIV testing.

## Figures and Tables

**Figure 1 ijerph-18-11252-f001:**
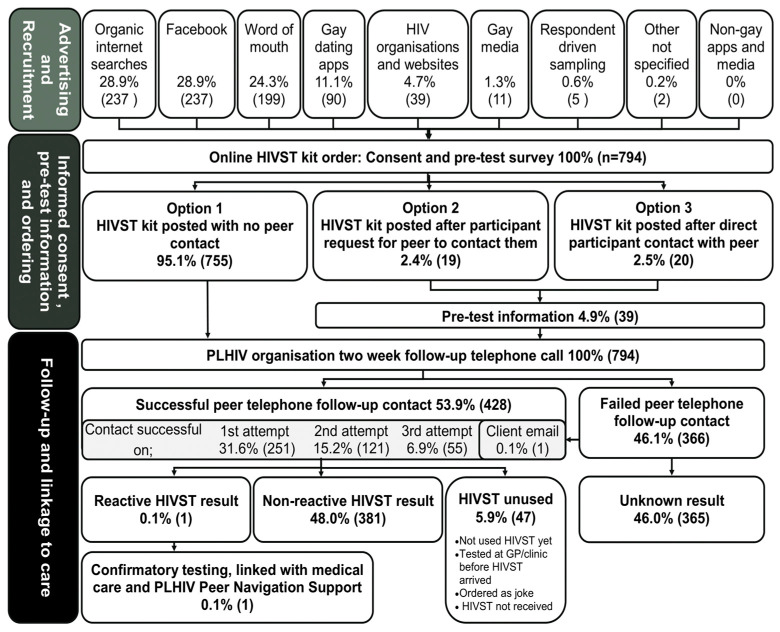
Schematic representation of the HIV self-testing study flow and outcomes of the first ordered HIVST kits (*n* = 794).

**Figure 2 ijerph-18-11252-f002:**
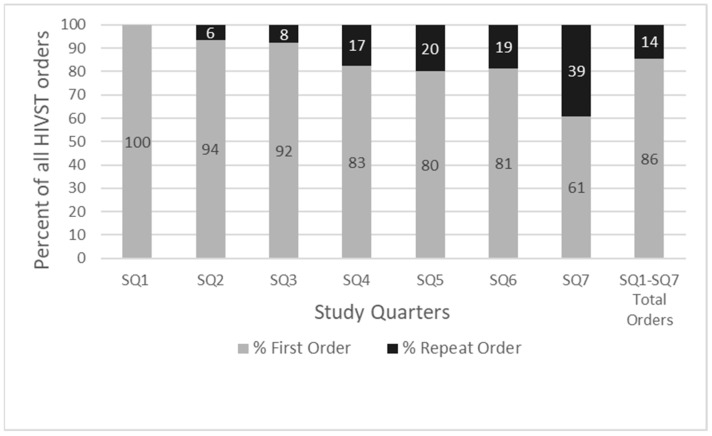
Percentages of first and repeat orders of HIVST kits ordered, by study quarter (SQ).

**Figure 3 ijerph-18-11252-f003:**
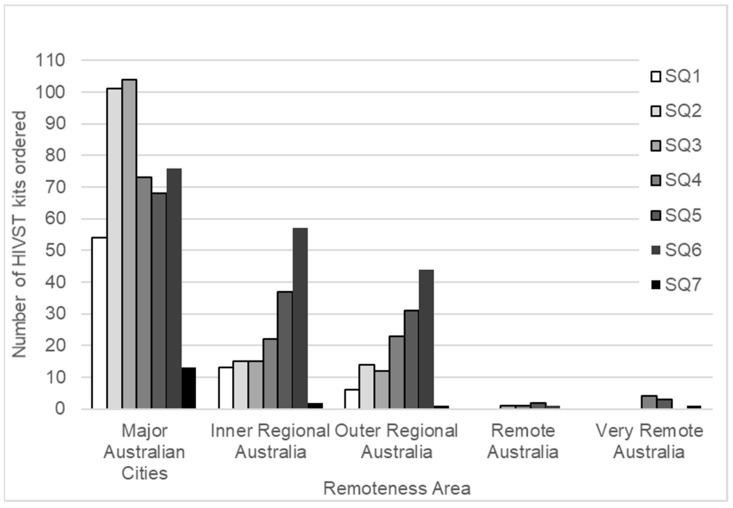
Number of HIVST kits ordered, by remoteness area and study quarter (SQ).

## Data Availability

As per the approved study protocol, access to these data is limited to named investigators and peer workers, and remains the property of the participating service providers. Access to these data may be considered by contacting the corresponding author of this manuscript. Access to the full study protocol is available in print [38].

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
