# Peer review of "Online HIV Self-Testing (HIVST) Dissemination by an Australian Community Peer HIV Organisation: A Scalable Way to Increase Access to Testing, Particularly for Suboptimal Testers"

_ijerph, 2021, doi:10.3390/ijerph182111252_

Round 1
Reviewer 1 Report
Thank-you for the opportunity to review this paper. I found it to be an interesting study, with well-written Discussion and Conclusions. I have no suggestions to make.
The paper looked at the characteristics of people using an HIV self-testing kit that was ordered at no cost from a community Peer-based HIV Organisation. Many of the recipients of the test had not tested previously, or had not tested within the last 12 months, but were in risk groups affected by HIV. The paper would be of interest to those working in the HIV testing field, and for those working in or with HIV community organisations.
The paper is reasonably original in that it is an HIV community organisation that has sent out the tests and arranged for follow-up with the recipients.
It is well-written and easy to follow with no spelling or grammar problems that I could detect.
The conclusions are relevant and consistent with the data presented from the study they conducted. The conclusions also make the point that cost may be a problem for people wanting to access such a test.
Author Response
Thank you for your comments and support of your work
Kind regards
Judith Dean on behalf of the team
Reviewer 2 Report
The article is a work that can contribute to awareness of the benefits of a remotely accessed HIV testing support. There are a couple items that should be addressed to enhance the final product.
- As the study is an evaluation study, the claims of its impact should be consistent. For example on line 450 the claim the organisation is uniquely prepared to address HIV is not part of this study design and should be modified or omitted. Line 446 moves to a claim that the study identified online access to HIVST, but in reality it was the only option evaluated and did not refute any other option as viable. Both seem readily fixable, and by scaling the outcomes to actual study it is an enhanced process and product.
- There is no description of the qualitative component and the results are unclear as to which component (quantitative/survey or qualitative) the results are attributable.
- In general keep the claims in line with the method and actual results.
Author Response
1. As the study is an evaluation study, the claims of its impact should be consistent. For example on line 450 the claim the organisation is uniquely prepared to address HIV is not part of this study design and should be modified or omitted. Line 446 moves to a claim that the study identified online access to HIVST, but in reality it was the only option evaluated and did not refute any other option as viable. Both seem readily fixable, and by scaling the outcomes to actual study it is an enhanced process and product. Response Thank you for your comments. The claims re impact have been revised and as suggested. Line 450 statement re unique position has been deleted and revised to read This well received model of HIVST distribution, implemented by a peer PLHIV community organisation with established links to HIV testing information, referral and support for people newly diagnosed with HIV uniquely positioned to address HIV stigma, can ameliorate psychosocial, financial, and geographical barriers to HIV testing whilst providing support for those wishing to test for HIV if desired. Line 446 has been revised to read The study identified online access to HIVST is a feasible and acceptable testing option for MSM in Queensland, Australia as an important addition to the existing suite of testing options in Queensland, particularly for engaging those who have never tested or who would not have tested through other methods.
2. There is no description of the qualitative component and the results are unclear as to which component (quantitative/survey or qualitative) the results are attributable. Response The following sentence has been added This paper presents the results of the quantitative online survey data collected pre and post dissemination of the kits. to clarify that the qualitative component is not reported in this paper.
3. In general keep the claims in line with the method and actual results Response The discussion and recommendations have been reviewed to align with the method and results (see track changes in document). |
Reviewer 3 Report
I’ve carefully revised your manuscript “
Online HIV Self-testing (HIVST) Dissemination by an Australian Community Peer HIV Organisation: a scalable way to increase access to testing particularly for suboptimal testers”.
Here my comments to improve your work:
- Please revise your abstract for effectiveness and remember your abstract should be a brief summary of the manuscript.
- Please revise your manuscript and the references for errors.
- Please make sure to include the strengths and the limitations for your study.
- Also please make sure to discuss what are the implications of your research in actual clinical practice.
Author Response
Thank you for your considered comments. Please find below our responses
Reviewer 3 |
Response |
1. Please revise your abstract for effectiveness and remember your abstract should be a brief summary of the manuscript.
|
The Abstract has been reviewed based on the reviewers suggestions. |
2. Please revise your manuscript and the references for errors.
|
Manuscript and reference list revised for grammar and errors |
3. Please make sure to include the strengths and the limitations for your study.
|
A strengths and limitations section has been added to the manuscript (Line 466 in track change document) |
4. Also please make sure to discuss what are the implications of your research in actual clinical practice. |
The manuscript has been reviewed and implications to clinical practice explicit reference to the implications for clinical practice have been noted throughout the Discussion, including in the Strengths and Limitations subsection. |